# Applying Multi-Purpose Commercial Inertial Sensors for Monitoring Equine Locomotion in Equestrian Training

**DOI:** 10.3390/s24248170

**Published:** 2024-12-21

**Authors:** Christina Fercher, Julia Bartsch, Steffen Kluge, Franziska Schneider, Anna M. Liedtke, Axel Schleichardt, Olaf Ueberschär

**Affiliations:** 1Olympic Training Center North-Rhine/Westphalia, 48231 Warendorf, Germany; christina.fercher@osp-westfalen.de; 2Department of Engineering and Industrial Design, Magdeburg-Stendal University of Applied Sciences, 39110 Magdeburg, Germany; julia.bartsch@h2.de (J.B.); franziska.schneider@h2.de (F.S.); 3Department of Mechanical Engineering, Chemnitz University of Technology, 09107 Chemnitz, Germany; 4Department for Veterinary Medicine, German Olympic Committee for Equestrian Sport, 48231 Warendorf, Germany; aliedtke@fn-dokr.de; 5Institute for Applied Training Science, 04109 Leipzig, Germany; schleichardt@iat.uni-leipzig.de

**Keywords:** limb loads, asymmetry, motion analysis, sport horses, injury prevention

## Abstract

Inappropriate, excessive, or overly strenuous training of sport horses can result in long-term injury, including the premature cessation of a horse’s sporting career. As a countermeasure, this study demonstrates the easy implementation of a biomechanical load monitoring system consisting of five commercial, multi-purpose inertial sensor units non-invasively attached to the horse’s distal limbs and trunk. From the data obtained, specific parameters for evaluating gait and limb loads are derived, providing the basis for objective exercise load management and successful injury prevention. Applied under routine in-the-field training conditions, our pilot study results show that tri-axial peak impact limb load increases progressively from walk to trot to canter, in analogy to stride frequency. While stance and swing phases shorten systematically with increasing riding speed across subjects, longitudinal and lateral load asymmetry are affected by gait at an individual level, revealing considerable variability between and within individual horses. This individualized, everyday approach facilitates gaining valuable insights into specific training effects and responses to changing environmental factors in competitive sport horses. It promises to be of great value in optimizing exercise management in equestrian sports to benefit animal welfare and long-term health in the future.

## 1. Introduction

Individual training monitoring is characterized by the systematic collection and analysis of data to oversee and optimize an athlete’s training. Its primary aim is to track the athlete’s performance development and health status to optimize training on an individual basis. In equestrian sports, efficient monitoring of the movement of elite horses presents numerous challenges. Besides the self-evident requirement for measurement accuracy, the monitoring systems must be easy to apply and must not interfere with the movement of the horse or rider to ensure compliance and continuous use during exercises. These systems also have to adapt to the widely changing environmental conditions of equestrian sports and account for the horse’s natural behaviour. Additionally, the significant differences in horses’ gait patterns must be considered. With four legs, horses exhibit multiple variations of natural gaits based on the sequence of footfalls [1,2]. The natural basic gaits of the horse—walk, trot, and canter—can be performed at varying speeds [3,4,5,6] and can additionally be extended to artificial gaits, such as passage and piaffe [7], or jumping strides in show jumping [8]. Being quadrupeds, horses distribute loads not only laterally between the left and right limbs while moving, but also longitudinally (i.e., along the craniocaudal axis) between the fore- and hindlimbs. Anatomically and functionally, the fore- and hindlimbs differ significantly. While the front legs naturally bear more of the body weight and act as shock absorbers, the hind legs are the main motors of propulsion [2,7]. Compared to bipeds, this difference results in significantly greater variation in movement patterns and compensatory adjustments [9,10], which are additionally influenced by locomotion speed, ground properties and surface conditions, track design, and shoeing [6,9,11]. This complexity must be adequately taken into account to enable a reliable, systematic monitoring of limb loads (health objective) and a valid assessment of the horse’s movement technique (performance objective) in elite sport training.

So far, significant advances in motion analysis for equestrian sports have been achieved through technological progress in recent decades. While force plates are used in the lab to determine limb loads and provide insights into load distribution between legs, they are difficult to implement in daily training [12,13]. Likewise, employing dynamometric horseshoes in the regular monitoring of limb loads is equally limited due to the instrument’s additional height and interference with the horse’s regular shoeing management [11]. Under field conditions, inertial measurement units (IMUs) have proven particularly useful [8,14,15]. With sensors positioned on the hooves [15,16], cannon bones [12,17], and the upper body line (consisting of the head, withers, and sacrum of the horse [1,18]), motor limb coordination in the gaits can be recorded and analysed, particularly through the support times of the limbs [19]. In addition to those temporal parameters of the stride cycle, the duty factor (the ratio of stance time to stride duration) can be used to assess gait symmetry [1,2,7] and to estimate the vertical ground reaction forces [18,20].

The aim of many studies focusing on equine locomotion is to provide technical support for diagnosing lameness. This complex task is traditionally carried out in trot and covers straight and curved sections on hard and soft ground surfaces to identify the affected leg. Examination results on a curved line on the lunge or under the rider may be helpful here [21]. Differentiating mild degrees of lameness from natural asymmetries in healthy horses proves particularly challenging [22,23,24]. For this reason, several studies have investigated the movement symmetry of sound horses and their natural adaptations to straight and curved lines or varying ground surfaces [25,26,27,28]. Notably, changing track surfaces and riding straight and curved lines with different radii are common scenarios in the daily training of sport horses. To allow for monitoring horses in their specific training context on a regular basis, a simple-to-use and self-sufficient technical instrument is required to reliably objectify the individual gait symmetry in the basic gaits on different riding surfaces. Various comparative studies have shown that the precise determination of hoof-on and hoof-off events in horses, which is needed for the accurate calculation of gait events, can be obtained using IMUs placed laterally on the horse’s cannon bones within the equine boot [12,13,14,17]. This approach appears to be most suitable for employment in the training of sport horses under everyday field conditions.

Motivated by the widespread successful employment of inexpensive, multi-purpose IMUs in human running monitoring [29], the objective of the present study was to evaluate whether such commercial multi-purpose IMUs can be employed for valid and specific monitoring of limb loads, symmetry, and possibly signs of fatigue in sport horses under field conditions and for different gaits. In particular, this study aimed to test whether tri-axial accelerometer and gyroscope data can be used to derive three-dimensional impact loads, their inter-limb distribution, and their spatiotemporal coordination to provide valid parameters for comprehensive training monitoring in equestrian sports. This approach has the potential to be transferable to technique analysis in dressage and jumping-specific training sessions, incremental exercise tests, and uphill canter work, as well as to health checks of sport horses.

## 2. Materials and Methods

### 2.1. Inertial Measurement Units (IMUs)

In this study, two types of commercially available, multi-purpose IMUs with identical sensor hardware specifications were used: Xsens MTw Awinda IMUs and Xsens DOT (both by Movella Inc., Enschede, The Netherlands). Each single IMU combines tri-axial data of 3D accelerometers (±16 g), 3D gyroscopes (±2000°/s), and 3D magnetometers (±800 μT) by sensor fusion using an internal data sampling rate of 1000 Hz (accelerometers and gyroscopes) and an output data rate of 120 Hz. The MTw Awinda IMUs have been used and validated for a wide range of motion analysis purposes in human sports science [30,31], and are also suitable for analysing equine motion [32,33,34,35]. Data transmission from the MTw Awinda IMUs to the receiving base station and synchronization among them were realized via a proprietary wireless protocol of the manufacturer using the 2.4 GHz band with a free space range of 50 m. The base station was set up on a small table within wireless range near the respective riding area or arena. The recently available IMU type DOT offers a low-cost alternative with the decisive advantage of being independent of radio transmission during the sessions. This allows for monitoring sessions under arbitrary field conditions and over much wider areas. These DOT sensors were operated in a synchronised but otherwise autonomous data logger mode. Both the MTw Awinda and the DOT sensors were initialized and prepared according to the manufacturer’s recommendations (i.e., proper temperature acclimatization period to adapt to outdoor conditions, sensor self-initialization in rest after switching on, sensor heading reset, etc.) before they were attached to the horse’s gear.

### 2.2. Sensor Positioning

Four IMUs of each sensor type were non-invasively attached to the horses’ four limbs on the sport-specific boots protecting the legs, positioned laterally at the cannon bone (Figure 1a,b). This positioning and attachment method are easy to implement [17] and well suited for measuring the horse’s stride characteristics and limb kinematics [14,33,36], as well as the compression load on the horse’s legs. The sensors were applied to the horse immediately prior to measurement. Care was taken to ensure that the sensors were always applied perpendicular to the axis of the cannon bone and by the same researcher in the same order: starting with the forelimbs, followed by the hindlimbs and eventually attaching the fifth sensor pair to the saddle. Covered with plush and Velcro tape, the four limb sensors allow quick attachment to the horse’s legs during training using the horse’s individual boots. To measure the trunk movement, the fifth pair of IMUs was attached to the horse’s saddle girth using another Velcro strap (Figure 1c). The positioning on the caudal part of the horse’s sternum via the saddle girth also proved to be robust and, due to its proximity to the body’s centre of gravity, it is particularly suitable for recording the trunk segment as the largest part of the horse’s mass [8,37]. When a stud girth was used, the fifth sensor pair was fixed to the right side of the saddle girth. The recorded trunk accelerations clearly reflected the rhythmic movement of the horse’s body over time when moving in different gaits. The corresponding rhythm is suitable for automated gait classification [38] and for detecting acyclic jumping movements [8,39,40]. After proper attachment, all sensors were activated and synchronized with the other sensors of their kind (i.e., either MTw Awinda or DOT) remotely.

Owing to different attachment sites, the coordinate systems of the five IMUs of each kind were aligned differently in the global coordinate system. The individual, local coordinate systems of the five IMU pairs are depicted in Figure 1c. However, for the later analysis of ground contacts, loads, and asymmetries in the limbs, only the magnitude of the 3D accelerations vectors (Euclidean norm) was used, i.e.,
(1)$$\left|\vec{a}\left(t\right)\right|=\sqrt{{\left(a_{x}\left(t\right)\right)}^{2}+{\left(a_{y}\left(t\right)\right)}^{2}+{\left(a_{z}\left(t\right)\right)}^{2}}$$

This value is invariant under coordinate system rotations, so that the exact spatial alignment of the sensor axes is irrelevant. To enable single-axis analysis for future purposes, however, the *x*-axis is always aligned parallel to the approximate cannon bone axis, with the positive direction pointing upwards, while the *z*-axis points medially (inwards). For the girth sensor, the *z*-axis points upwards and the *x*-axis in a forward direction (Figure 1c).

Step cycle segmentation was accomplished for each sensor type and sensor independently by using the periodic shape of its global-frame pitch angle, i.e., the inclination with respect to the global horizontal reference plane defined by Earth’s gravity perpendicular to it and pointing downwards (corresponding to a positive, upward acceleration reading of the IMU of +9.81 ms^−2^).

### 2.3. Study Design

The study was integrated into the daily training routines of various sport horses. The primary gaits of walk, trot, and canter were recorded for a total of 20 horses using the previously described measurement techniques and positioning. The objective was to analyse inter-individual variability in limb coordination through stride frequency, duration of the stance and swing phases, limb loads via peak impact accelerations, and their distribution across the four limbs in different gaits. The cohort size was chosen to ensure sufficient group sample sizes in subsequent repeated-measures analysis of variance (ANOVA) [41]. For each gait, four trials of about 40 s each were recorded, with half of the trials performed on the left rein and the other half on the right rein. The canter’s asymmetric stride pattern can be differentiated into right-lead and left-lead canter. Therefore, two trials were conducted on the left rein in the left-lead canter, while the other two were on the right rein in the right-lead canter. To illustrate the gait patterns, Figure 2, Figure 3 and Figure 4 exemplify the general movement phases of the walk (Figure 2; 3 beats with 6 phases), trot (Figure 3; 2 beats with 4 phases), and canter (Figure 4; 3 beats with 6 phases) with the footfalls during stance phases and important acceleration and angular velocity landmarks for one horse. Figure 4 shows a sequence of a right-lead canter of one horse, where the right limbs lead the canter stride.

The data collection sequence always started with a walk, followed by a trot, and ended with a canter. The rider determined the order of direction and speed, following the instruction that it should correspond to the horse’s normal working pace in the respective gait. The posture of the horses and the rider’s sitting position were chosen by the rider. The horses were trained in a familiar riding arena with a sand-mixture surface. The trials included both straight and curved segments within a rectangular path of approximately 25 m × 50 m, ensuring a consistent proportion of straight and curved paths for all horses. These short gait intervals generally correspond to excerpts from a normal training session.

The horses were individually prepared by the riders and were granted sufficient time to adjust to their environment. The exercises and the duration of the individual trials, as well as the overall extent of the data collection, did not pose any form of unfamiliar challenge for the participating horses. No unusual or unnatural movements or tasks were expected from the riders or horses. During the entire data collection procedure, a veterinarian observed the movement of the horses to oversee their well-being and to detect any sign of emerging lameness as an exclusion criterion.

The entire process of data acquisition was accompanied by a video camera (Sony HXR-NX5E, 50 fps, Full HD) to facilitate later analysis and validate kinematic results. Sensor–video synchronization was carried out mechano-visually by tapping on the sensor at the horse’s trunk, clearly visible in the video sequence at the beginning and at the end of each measurement trial. Eventually, sensor positioning and data completeness were checked after each trial. Given the different acquisition rates of inertial data of 120 Hz and video images of 50 fps, the maximum measurement uncertainty of detecting spatiotemporal events during methodological validation is 15 ms.

### 2.4. Horses

A total of 20 different horses and their riders participated in our pilot study. The group consisted of 11 geldings, 8 mares, and 1 stallion. The horses ranged in age from 4 to 14 years and represented various disciplines of equestrian sports, including dressage (*n* = 10), show jumping (*n* = 4), and eventing (*n* = 6), to obtain a wide range of motion data for a comprehensive methodological analysis. They exhibited performance levels corresponding to their ages and stages of training. Inclusion criteria were healthiness, in particular showing no signs of lameness (i.e., being sound sport horses in training to compete), the interest and willingness of their rider to participate in this study, and the acceptance of the horse to wear protective boots on all four legs. The horses were presented by their daily rider in a typical riding arena for data collection during training.

### 2.5. Data Analysis

For the MTw Awinda sensors, the IMU data were recorded using the manufacturer’s MT Manager 4.7 software suite (Movella Inc., Enschede, The Netherlands). Data collection with the DOT sensors was started and stopped using the manufacturer’s Android app DOT on an Android smartphone (Samsung Galaxy S21 5G Enterprise, Samsung Electronics Co., Ltd., Suwon, Republic of Korea). The data were then exported and analysed using a custom-made MatLab script (R2022b, MathWorks Inc., Natick, MA, USA). Gait variables were analysed by automatically extracting all gait cycles based on the cyclic sensor orientation and peak accelerations. For gait cycle segmentation, the cycle duration of each stride was first estimated by analysing separately the oscillating pitch angles of all five IMUs of a sensor type in the global reference frame, with the pitch angle describing the oscillating inclination from the horizontal plane (Figure 5). This quantity proved stable and best suited among all three Euler angles, acceleration, and gyroscope axes for cycle detection in all gaits. Based on this cycle period, the corresponding peaks in tri-axial acceleration magnitudes during ground contact phases were detected (Figure 2, Figure 3 and Figure 4). Based on this cycle period, the corresponding peaks of the angular velocity in the sagittal plane were determined using the method developed by Sapone et al. [17].

The signal was pre-processed with a second-order Butterworth low-pass filter at 20 Hz. The period between the identified peaks of the filtered signal defines a complete gait cycle, beginning in the swing phase. Based on the work by Briggs et al. [14], the first peak within this gait cycle in the tri-axial acceleration marks the moment of initial hoof contact with the ground (hoof-on), while the second prominent peak indicates the moment of hoof lift-off from the ground (hoof-off). This methodology was shown to be highly reliable in walk and trot under field conditions with different surfaces [14]. Figure 5 shows the data using a time series at a trot as an example. In this study, Briggs’ approach is extended also to canter.

With the hoof-on and hoof-off events of the individual limbs, a variety of biomechanical parameters can be derived that describe and evaluate the individual gait pattern of a horse [2,9,13,20]. For this primarily methodological and practice-centred pilot study, the following parameters were selected and evaluated in the basic gaits of walk, trot, left canter, and right canter: stride frequency (i.e., inverse cycle period), duration of swing and stance phase of each limb, and peak impact accelerations as surrogate measures of the compression loads in the horses’ cannon bones. Moreover, the asymmetry of load distribution between the fore- (*F*) and hindlimbs (*H*), as defined by the longitudinal asymmetry index (*LongAI*),
(2)$$LongAI:=\frac{F-H}{\frac{1}{2}\left(F+H\right)}\cdot 100\%$$
as well as the asymmetry of load distribution between the left (*L*) and right (*R*) limbs, as given by the lateral asymmetry index (*LatAI*),
(3)$$LatAI:=\frac{L-R}{\frac{1}{2}\left(L+R\right)}\cdot 100\%$$
were calculated to observe the symmetry of the novel compression limb loads of horses.

### 2.6. Statistical Analysis

All statistical analyses were conducted using the open-source software suite JASP (Version 0.19, JASP Team, Amsterdam, The Netherlands). Results are presented as mean values with corresponding standard deviations (SD). To evaluate gait- and direction-dependent changes in the parameters, a repeated-measures ANOVA was performed. Post hoc comparisons following the ANOVA were conducted using Tukey’s test. Effect sizes for the ANOVA are reported as partial *η*^2^, while Cohen’s *d* is used for post hoc analysis. Effect sizes are categorized as small (0.2 ≤ *d* < 0.5 or 0.01 ≤ *η*^2^ < 0.06), medium (0.5 ≤ *d* < 0.8 or 0.06 ≤ *η*^2^ < 0.14), and large (*d* ≥ 0.8 or *η*^2^ ≥ 0.14) [42]. Normal distribution was confirmed using Shapiro–Wilk tests, while homoscedasticity was assessed with Levene’s test. The significance level was set at α = 0.05. The relationships between parameters are analysed using Spearman’s correlation coefficients.

## 3. Results

Comparing the results for the two sensor types, Xsens MTw Awinda and Xsens DOT, in terms of the detected hoof-on tri-axial peak impact acceleration magnitudes across all three gaits, a very strong correlation and good agreement were found (Figure 6, sample Pearson correlation coefficient of *r* = 0.990, *R*^2^ = 0.980, *p* < 0.0001; Bland–Altman mean deviation of 0.38 g ≙ 3.7%). Given this practical equivalence of both sensor types, the following analyses will focus on the MTw Awinda sensors, owing to their well-established, widespread use in sports science.

The analysis of stride duration and limb loads, as well as their lateral and longitudinal asymmetries across different gaits, provides valuable insights into the movement of sport horses. Table 1 presents the parameters for stride frequency, stride duration, and the tri-axial peak impact acceleration defining peak impact limb load (PILL) in walk, trot, and canter.

The influence of the gaits on the individual limb parameters is clearly evident and illustrated in Figure 7. Stride frequency significantly increases from walk to trot to canter (*p* < 0.001; *η*^2^ = 0.989). Stride duration, encompassing both stance and swing phases, also varies with gait, showing a decrease from 1.21 s in walk to 0.63 s in canter. Similarly, stance phase duration significantly changes across gaits (*p* < 0.001; *η*^2^ = 0.987). As for swing phase duration, ANOVA reveals significant differences between gaits (*p* < 0.001; *η*^2^ = 0.793), with an increase in swing duration from walk to trot (*Cohen’s d* = −0.48; *p* = 0.006) and a significant decrease in canter (*Cohen’s d* = 3.57; *p* < 0.001). From walk to canter, there is also a significant reduction in the total stride duration. PILL increases with gait, rising from 6.55 ± 1.28 g in walk to 9.77 ± 1.04 g in trot and to 14.03 ± 0.93 g in canter (*p* < 0.001; *η*^2^ = 0.939). Canter is associated with the highest movement frequency, shortest swing and stance duration, and greatest PILL. It can thus be concluded that this gait is associated with the greatest strain on the musculoskeletal system.

The longitudinal and lateral load distribution of the compression limb load is shown in terms of asymmetry indices *LongAI* and *LatAI* in Table 2. Longitudinal load distributions between the fore- and the hindlimbs show a significant dependence on gait (*p* < 0.001; *η*^2^ = 0.100). Post hoc tests provide further differentiation of the results: In trot, the load tends to shift towards the hindlimbs. In walk, the load distribution is still on the hindlimbs, but slightly more balanced than in trot. The load distribution asymmetry in canter is less than 1% and thus virtually balanced. Moreover, walk does not significantly differ from the other two gaits in terms of longitudinal asymmetry, being the middle piece between the *LongAI* of trot and canter (*p >* 0.05, *Cohen’s d* = −0.28–+0.27), whereas trot and canter show a significant difference between each other (*p* < 0.001; Cohen’s *d* = −0.551). Figure 8a depicts the mean longitudinal compression limb load distribution in the gaits.

In addition to gait, the individuality of each horse influences *LongAI*, as shown in Figure 8b. The intra-individual variance in longitudinal load distribution is high, as illustrated by the box plots, with symbols indicating the mean values for each gait of the trial. Some individual horses exhibit a consistent and pronounced asymmetry in load distribution: While horses 2, 3, 12, and 15 tend to have a dominance towards the forelimbs, horses 6, 7, 14, and 17 show a slightly higher compression load tendency towards the hindlimbs. There seems to be no relation to the horse’s equestrian discipline in the cohort studied. The lateral asymmetry *LatAI* of the peak compression loads was analysed separately for the fore- and hindlimb pairs. Overall, gait does not significantly affect *LatAI* for either the fore- or hindlimbs (*p* > 0.05).

Riding direction, along with the associated turns, can also impact the compression load distribution across the limbs and was thus studied. While riding direction does not affect the longitudinal load distribution *LongAI* and shows no interaction effects with gait (*p* > 0.05), the situation is different for lateral load distribution *LatAI*. Figure 9 illustrates the lateral distribution of PILL between the left and right limbs as a function of gait and riding direction. *LatAI* is significantly influenced by riding direction in the forelimbs (*p* = 0.001; *η*^2^ = 0.041), but not in the hindlimbs (*p* = 0.124; *η*^2^ = 0.009). The interaction between gait and direction shows a significant effect for both fore-(*p* = 0.016; *η*^2^ = 0.046 = and hindlimbs (*p* < 0.001; *η*^2^ = 0.237). In contrast, no significant side differences are observed between the left and right forelimb during walk and canter (*p* > 0.05). However, in trot, the lateral load distribution of the front limbs is counter-lateral to the riding direction—right trot favours the left limb whereas left trot favours the right limb (*p* < 0.001; *d* = 0.835). The hindlimbs show a significant effect between the gait direction in trot (*p* = 0.009; *d* = −0.924) and also in canter (*p* < 0.001; *d* = −1.524). In the asymmetrical footfall pattern of canter, the riding direction causes a slight lateral shift toward the outer hindlimb (9.5–12.0%). Conversely, in the symmetrical foot sequence of trot, the riding direction results in a minimal shift toward the inner hindlimb (6–7%).

Overall, the average asymmetry of load distribution among the horses’ four limbs is slight, with mean values predominantly below 10%. The distribution of the high loads on the front legs remains almost balanced across all movement cycles in the 20 cantering horses, while the hindlimbs show a slight asymmetry, with a shift towards the outside. In trot, the hindlimbs adapt more to the riding direction with lateral distribution shifting inwards, while the forelimbs shift counter-laterally to the outside. Walk, as a symmetric gait without a suspension phase, has the smallest compression limb loads and appears to be the most independent in terms of riding direction and the most laterally and longitudinally balanced gait. Detailed descriptive data are provided in Table 2. A pairwise comparison of 7704 strides of the four legs of 20 horses at walk, trot, and canter shows that the PILL correlates strongly with the support phase duration (*ρ* = +0.72–+0.74, *p* < 0.001).

## 4. Discussion

In summary, our study results demonstrate that the proposed measurement system based on commercial multi-purpose IMUs, along with the specific sensor placement, is well-suited for providing valid and specific monitoring of limb loads and symmetry in sport horses under field conditions and for different gaits. While the gold standard for measuring ground reaction forces, stance times, and load distribution involves the use of force plates [2,18,20], this method is difficult to implement under field conditions. In contrast, IMUs are considered highly suitable for biomechanical measurements under field conditions [13,18,20]. The developed system based on inexpensive, commercially available multi-purpose inertial sensors can be seamlessly integrated into daily exercise sessions and is fully compatible with the varying external factors in equestrian sports. The Xsens IMU monitors by Movella used in this study have been validated in various research fields for both humans and animals [30,31,33,34], ensuring scientific standards in terms of validity, reliability, and objectivity for these data. These IMUs are small, lightweight, and wireless, making the technology suitable for mobile use at different training locations.

Sensor placement is crucial for IMU data analysis and must be tailored to the specific needs of the particular study objectives. The current literature suggests that placing IMUs on the cannon bones of horses is a suitable approach, as it offers the advantage of securing the IMUs using specialized boots, in addition to positioning them near the hooves [12,18,43]. Although it seems intuitive to attach IMUs directly to the hooves due to their proximity to the ground [15,16], this method poses challenges, such as a higher risk of IMUs falling off or being damaged and the nature of the sensors being directly attached to the hoof wall [12,18]. Comparative studies have shown that results from cannon bone placement are reliable for estimating stride duration and stance times [1,12,14,17]. Moreover, the cannon bone placement of the sensors allows quick and easy attachment to the horse’s boots with Velcro tape directly before the start of an exercise session. Securing the sensors to the horse’s existing equipment while avoiding direct contact with the horse’s body increases acceptance among riders and trainers and is also ethically preferable.

Regarding strategies for analysing raw IMU data from cannon bone IMUs, there are several procedures described in the recent literature. While Darbandi et al. [12] studied algorithms for canter along with walk and trot, Hatrisse et al. [13] compared results for straight and curved lines. In addition to threshold analysis of tri-axial acceleration, uniaxial acceleration along the vertical axis of the cannon bone or angular speeds may be used to detect hoof events. Recently, new algorithms based on pattern recognition and artificial intelligence have been developed. However, due to the damping properties of the three joints more distal to the sensor position, a temporal delay in event detection is generally observed for cannon bone positions [12,14,17]. Following Sapone et al. [17] and Briggs et al. [14], we chose a combination of phase segmentation based on angular speeds and peak detection in tri-axial accelerations for our data analysis. This approach, which has been validated under both laboratory and field conditions, ensures a wide range of application scenarios and allows for comparison across different gaits and ground conditions. Focusing on limb coordination and rhythm through the symmetry of stance phases, gait purity and deviations due to functional limitations or lameness can also be assessed [2,12,13,14,17,43]: By analysing the timing of hoof contacts of the four legs, the gaits can be clearly differentiated, and the horse’s gait quality can be validly evaluated.

Considering load distribution, unlike bipeds, analysing quadrupeds such as horses requires considering both lateral and craniocaudal aspects. Tri-axial peak impact accelerations provide a detailed view of the loads on the distal limb segments, which is one of the most common injury localizations due to excessive or incorrect loading. At this stage, anatomical and functional differences between the biomechanics of the horse’s forelimbs and hindlimbs [2,7] cannot yet be incorporated into the IMU data analysis, while damping by the hoof and fetlock joint is accounted for. Nevertheless, attaching sensors along the longitudinal axis of the distal limb segment enables comparable measurements of impact loads on fore- and hindlimbs. Future research should further investigate the role of anatomical and biomechanical differences between the fore- and hindlimbs in IMU readings.

Analysing movement data from 20 sport horses revealed that stance duration, stride frequency, and peak impact limb loads progressively increase from walk to trot to canter. Significant variations in limb loads were observed both between and within individual horses, with notable differences in load distribution across strides within the same gait. The gait intervals, conducted at a standard working pace, included both straight and curved sections, as typically seen in training, which may contribute to this variability (as well as factors such as speed and ground surface; see below) [7,13]. The results suggest that both gait and direction significantly influence the distribution of loads. Walk, characterized by its four beats and six phases without a suspension phase, is the most balanced gait and places the least stress on the four limbs. In canter, a 3-beat, 6-phase gait, tri-axial peak impact loads are significantly higher compared to walk and trot. The distribution of the loads between the fore- and hindlimbs is balanced. In trot, a 2-beat, 4-phase gait, however, a slight shift in mean acceleration peaks towards the hindlimbs is observed. Noteworthily, differences in age and training level of the horses may limit the generalizability of these results.

The centre of mass of the horse, naturally located closer to the forelimbs in the trunk, typically results in a slightly higher static loading on the forelimbs [9]. Also, given the forelimbs’ functional role in shock absorption, higher peak impact limb loads are expected in the forelimbs. The hindlimbs, on the other hand, are responsible for propulsion. Training horses to be healthy riding horses aims to shift the centre of mass slightly backwards during movement by increasing hindlimb engagement [7]. Increased flexion and movement of the hindlimbs under the body towards the front lead to improved balance under the saddle (collection) and relieve pressure on the forelimbs. Additional kinematic data on other parts of the upper body, such as the withers, sacrum, and spine [32,33,36,44], can provide further valuable insights into the horse’s gait. Trunk rotation around the mediolateral axis, i.e., the “pitch angle”, reflects the uphill tendency of the horse, while the “roll angle” around the longitudinal axis can be used to determine the body lean angle. We therefore attached an additional sensor at each horse’s sternum, which was positioned on the horse’s girth using Velcro straps [24]. In future studies, this method could be easy to implement and potentially provide essential data on trunk movement, with this sensor capturing the trunk´s vertical oscillation and its rotation around the body’s transverse axis during each stride.

From the lateral perspective, the theoretical optimum would be a completely balanced distribution of peak loads across all four limbs of the horse in all gaits on a straight line [22,24]. For horses, natural crookedness and functional motor laterality play a crucial role in this context. Like other mammal species, horses have a dominant side, which influences their flexibility and balance and can lead to a natural asymmetric load distribution during locomotion. Therefore, many owner-judged sound horses show gait asymmetries on a straight path [23,27]. Additionally, horses lean their body to the inside when they are moving on a circular path [25,26], and adopt an asymmetrical movement pattern for balance [24,27]. In our study, the average lateral load distribution among the 20 horses was roughly symmetrical, with mean asymmetries of under 12%, including straight and curved path sections. There are several different grading scales for the standardization of the severity of lameness, commonly assessing the movements of the poll, the tubera coxae, and tubera sacrale at the upper line of the horse [21,22,24]. To date, there is no grading scale for natural asymmetry-based limb loads measured with IMUs on the cannon bones of horses, which limits comparability. Generally, it should be kept in mind that in any quadrupedal (and bipedal) vertebrae species, a lateral asymmetry may also simply reflect an effective adaptation to compensate for individual anatomical and/or orthopaedic conditions, and thus may not necessarily represent an imbalance. For reference, lateral asymmetries of up to 24% are reported to be “normal” in sound human elite runners [29]. With the monitoring system presented, an individual baseline can be created for each athlete during training, regardless of their main discipline, which can be documented and analysed throughout the entire training process itself and throughout the season.

As for gait differences, results for the per se symmetrical gait of trot indicate a tendency for load shifts towards the outer front limb and inner hindlimb, depending on the direction of riding. In the per se asymmetrical canter, in contrast, the load on the front limbs remains roughly balanced, while the load on the hindlimbs tends to shift outwards. These findings align with the results reported by previous studies. When moving on curved lines, horses tend to bend and lean inward [25], with variations in the body lean angle depending on their age and level of training [26], potentially reflecting differences in balance. This adaptation introduces a natural asymmetry in their movement patterns. Specifically, on the lunge, the stance phases are longer for the inner front limb and the outer hindlimb [24], which reduces the peak load on these limbs. Consequently, the peak load distribution inversely shifts to the outer front limb and the inner hindlimb. No significant differences between trot and canter were observed [26,28], presumably because the data encompassed movement cycles on both straight and curved paths, which may rebalance the load distribution in the forelimbs during canter. To gain deeper insights into limb load symmetry, further isolated studies should analyse peak impact limb loads separately on straight paths and curved lines with varying radii. This would enable a more detailed examination of the effects of different movement patterns.

In general, typical horse training sessions, whether in an arena, on an outdoor riding ground, in a forest, or on a racetrack, involve several transitions between straight and curved paths, while regular changes in riding direction promote an even distribution of limb loads. Future research on well-trained, adult horses performing standardized movement exercises should further elucidate those factors associated with symmetry. Importantly, individual analysis may generally provide valuable insights that may be blurred or lost in the average values presented. As for riding speed and ground conditions, all trials of this study were conducted on well-maintained sand surfaces, with four different riding arenas included in the study. Riders were instructed to maintain a consistent and regular working pace in each gait and to avoid excessive speed changes within a gait [3,4,5]. However, horse speed and ground parameters could not be objectively measured in the present study. Future research should thus aim to include these factors as further controllable variables. In such future studies, the advantage of virtually unlimited measurement spaces due to the absence of radio range limitations in the DOT sensors may prove highly beneficial.

In essence, the presented monitoring method provides valuable insights into the individual movement patterns of a horse. Evaluating the lateral and craniocaudal peak impact limb loads and their symmetry offers an objective measure of monitoring biomechanical loading during regular training. In fact, the long-term objective governing this pilot study is the establishment of an IMU-based measurement system for monitoring sport horses during their daily training routines. The monitoring system is intended to be practical for use under real-world field conditions while providing detailed insights into gait patterns and load distribution across various gaits and training modalities. It has the potential to be applied to further, more complex training scenarios in equestrian sports, e.g., documenting and analysing the rate of change of individual parameters during a specific dressage exercise or the foot sequence during take-off and landing phases in jumping sessions. By considering both preventive (health) and athletic (performance) aspects, this system may contribute to improved long-term training management for elite equine athletes by enhancing our understanding of individual responses to different training conditions and enabling us to tailor future training more effectively to the individual needs of each horse. Moreover, including individual movement data of the rider may open additional perspectives.

## 5. Conclusions

This study demonstrates that multi-purpose commercial inertial measurement units can readily be employed for monitoring horse locomotion under field conditions. The positioning and attachment of the sensors on the horse’s cannon bones using equine boots and on the horse’s trunk by means of the saddle girth prove practically applicable for everyday use. A segmentation and extremum detection algorithm based on combined accelerometer and gyroscope data from all five sensors yields valid results for all three basic equine gaits of walk, trot, and canter. The proposed, comparably inexpensive monitoring system allows for the individual evaluation of gait patterns and exercise loads to identify training effects and to detect individual reactions to specific external ground and environmental conditions, as well as to prevent injuries.

## Figures and Tables

**Figure 1 sensors-24-08170-f001:**
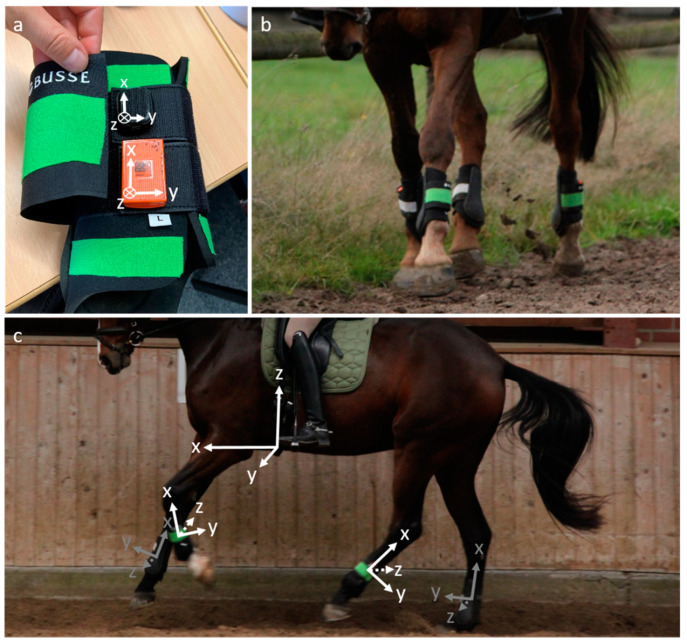
Attachment and positioning of the IMUs on the horse. (**a**) IMU attachment wallet for the boots with the Xsens DOT sensor (black) in the upper part of the wallet and the Xsens MTw Awinda sensor (orange) on the lower part; (**b**) placement on the cannon bones; (**c**) alignment of the local coordinate systems of each sensor pair.

**Figure 2 sensors-24-08170-f002:**
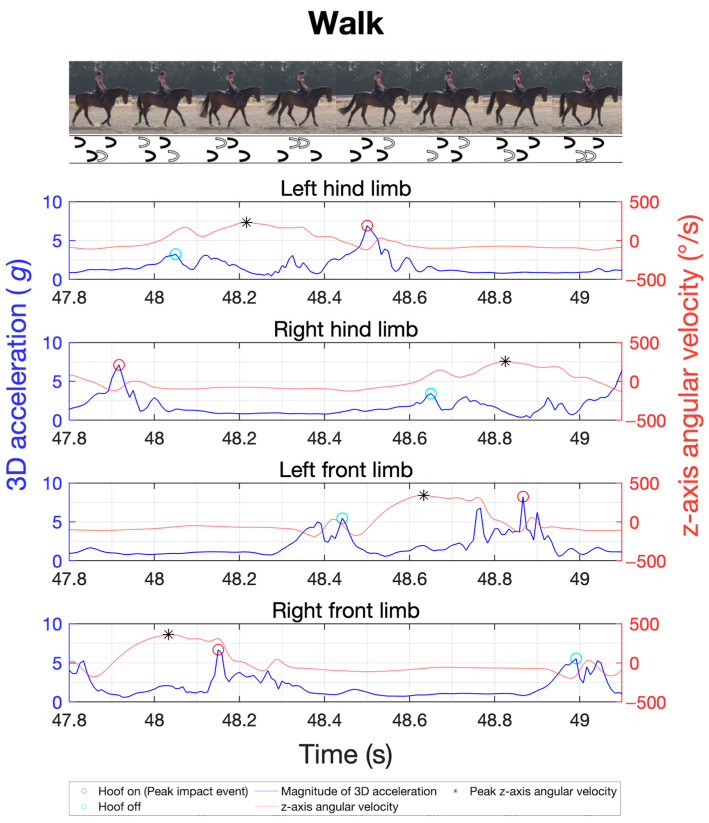
Sample of a time series of the tri-axial acceleration of the four limbs (blue) as well as the corresponding angular velocity (orange) of the sagittal plane for gait analysis in walk, as obtained with the MTw Awinda sensors. Red circles indicate the hoof-on and cyan circles the hoof-off event. The black asterisks represent the start and end of a gait cycle, as highlighted in grey in the image on the right.

**Figure 3 sensors-24-08170-f003:**
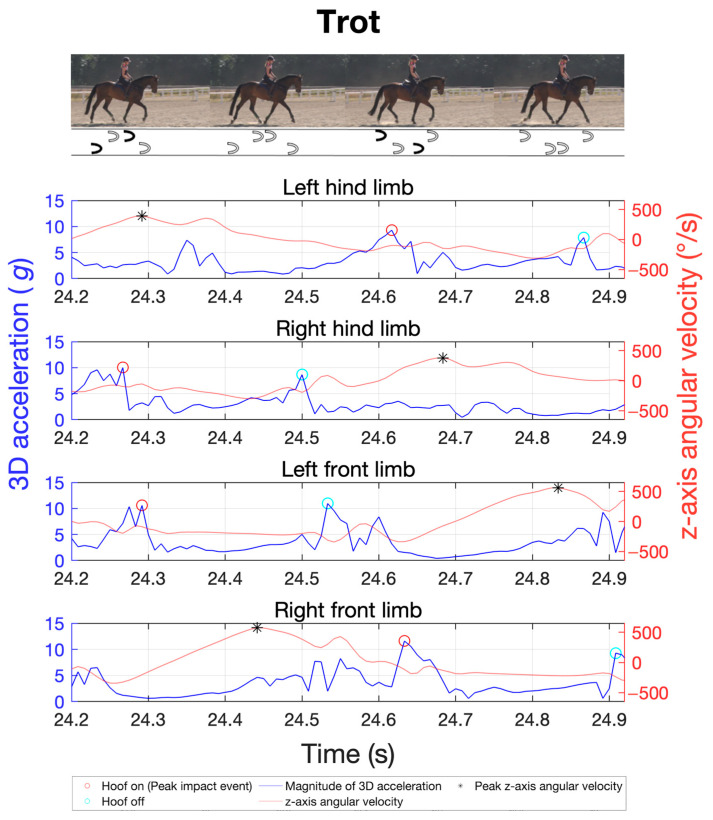
Sample of a time series of the tri-axial acceleration of the four limbs (blue) as well as the corresponding angular velocity (orange) of the sagittal plane for gait analysis in trot, as obtained with the MTw Awinda sensors. Red circles indicate the hoof-on and cyan circles the hoof-off event. The black asterisks represent the start and end of a gait cycle, as highlighted in grey in the image on the right.

**Figure 4 sensors-24-08170-f004:**
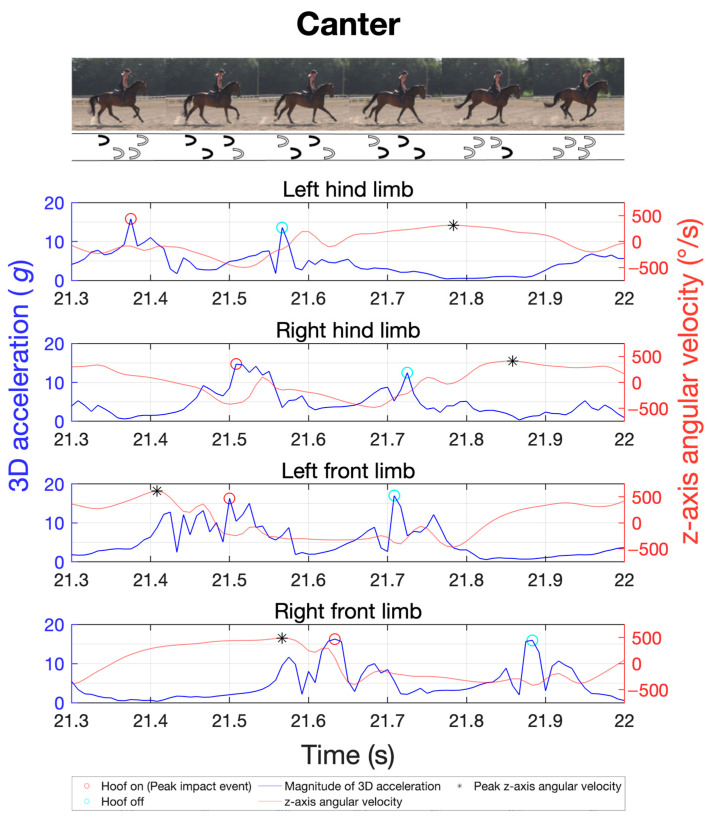
Sample of a time series of the tri-axial acceleration of the four limbs (blue) and the corresponding angular velocity (orange) of the sagittal plane for gait analysis in a right-lead canter on a straight line walk as obtained with the MTw Awinda sensors. Red circles indicate the hoof-on and cyan circles the hoof-off event. The black asterisks represent the start and end of a gait cycle, as highlighted in grey in the image on the right.

**Figure 5 sensors-24-08170-f005:**
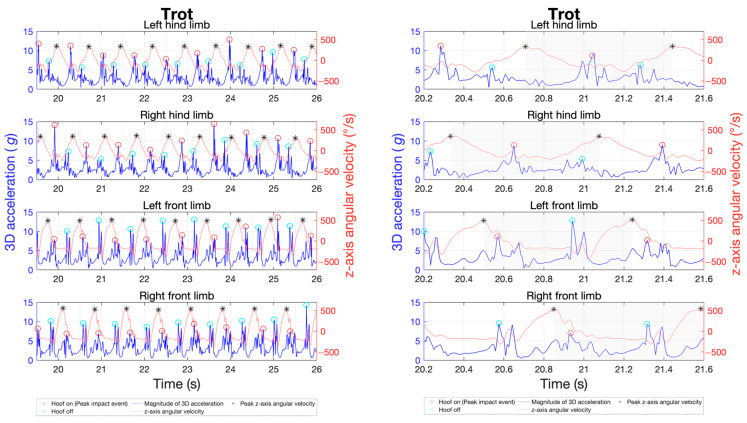
Sample of a time series of the tri-axial acceleration of the four limbs (blue) as well as the corresponding angular velocity (orange) of the sagittal plane for gait analysis (MTw Awinda sensors). Red circles indicate the hoof-on and cyan circles the hoof-off event. The black asterisks represent the start and end of a gait cycle, as highlighted in grey in the image on the right.

**Figure 6 sensors-24-08170-f006:**
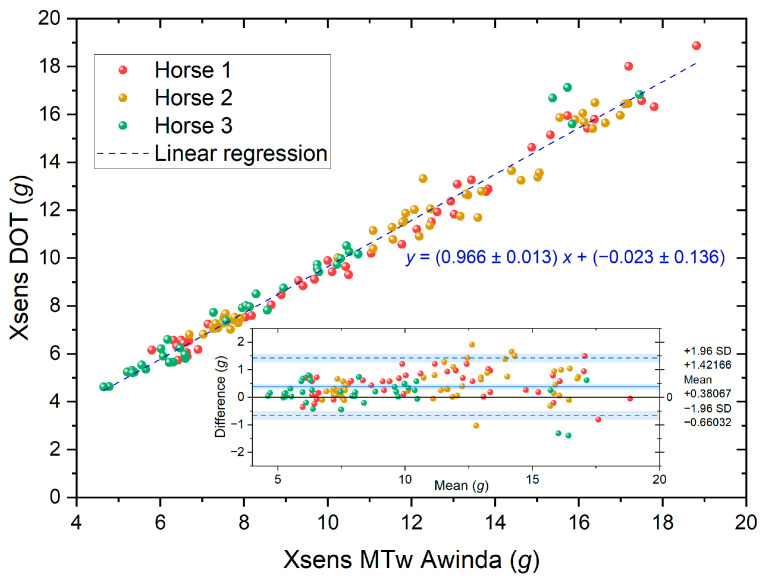
Comparison of results of the sensor types Xsens MTw Awinda (abscissa) vs. Xsens DOT (ordinate) for tri-axial peak impact acceleration magnitudes during walk, trot, and canter, as exemplarily sampled for three horses (colours). The inset shows the corresponding Bland–Altman plot.

**Figure 7 sensors-24-08170-f007:**
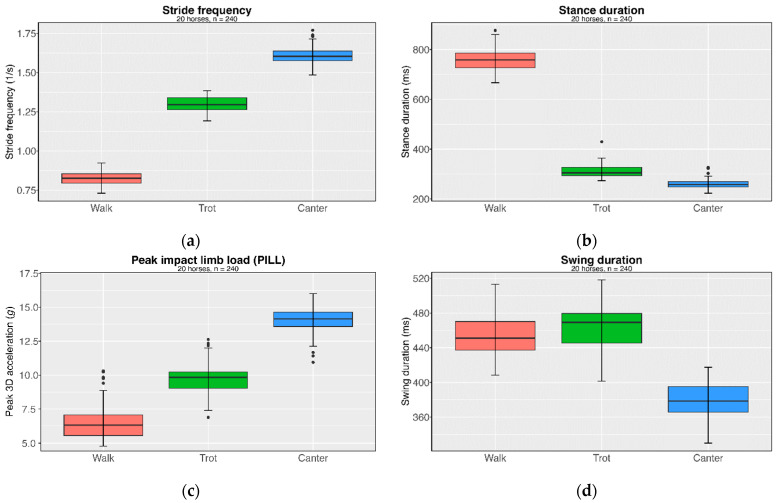
Cohort means and distributions of (**a**) stride frequency, (**b**) stance duration, (**c**) PILL, and (**d**) swing duration across different gaits for the cohort of 20 horses.

**Figure 8 sensors-24-08170-f008:**
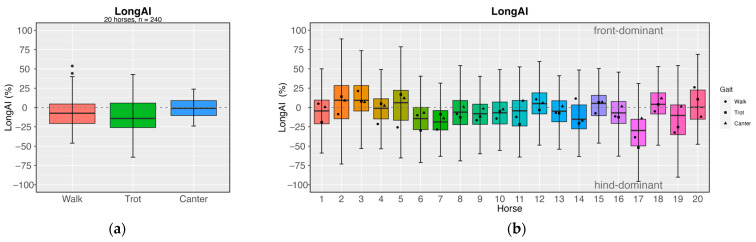
Cohort means and distribution of (**a**) cohort longitudinal index *(LongAI*) and (**b**) individual *LongAI* for the cohort of 20 horses. The 20 different colours in (**b**) from red to magenta illustrate the 20 individual horses.

**Figure 9 sensors-24-08170-f009:**
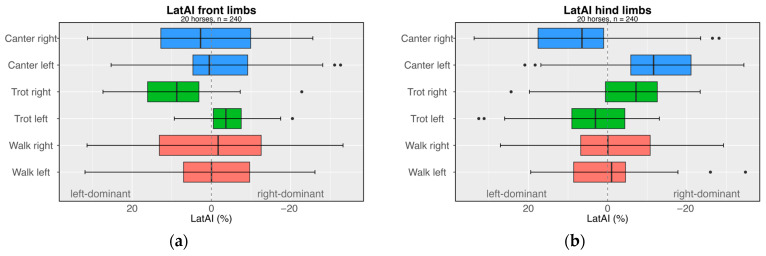
Mean lateral asymmetry index (*LatAI*) of the (**a**) forelimbs and (**b**) hindlimbs in terms of gait for the cohort of 20 horses.

**Table 1 sensors-24-08170-t001:** Mean stride frequencies, stride durations, and tri-axial peak impact acceleration (PILL) as a function of gait.

	Walk		Trot		Canter	
	Mean	±SD	Mean	±SD	Mean	±SD
Number of trials	80		80		80	
Stride frequency (1/s)	0.83	0.05	1.30	0.05	1.61	0.06
Stride duration (s)	1.21	0.07	0.78	0.03	0.63	0.02
Stance duration (ms)	757.69	49.70	313.18	26.87	260.46	20.80
Swing duration (ms)	454.09	24.47	463.60	25.90	378.53	19.53
Peak impact limb load (PILL) (g)	6.55	1.28	9.77	1.04	14.03	0.93

**Table 2 sensors-24-08170-t002:** Mean compression load distributions between fore- and hindlimbs (*LongAI*) as well as for right and left limbs (*LatAI*) as a function of gait.

	Walk		Trot		Canter	
	Mean	±SD	Mean	±SD	Mean	±SD
Number of trials	80		80		80	
*LongAI* (%)	−6.24	20.99	−11.53	23.87	−0.72	12.13
*LatAI*						
Number of trials	40/40		40/40		40/40	
Forelimbs (%)	1.20/−0.40	16.22/16.71	−4.10/7.67	6.60/12.02	−2.54/3.07	14.73/15.71
Hindlimbs (%)	−0.74/−0.67	11.36/12.43	6.15/−6.91	15.04/14.84	−12.04/9.51	13.44/17.01

Notes: A positive *LongAI* represents a higher load on the forelimbs, whereas negative values reflect higher loads on the hindlimbs. For *LatAI*, positive values denote a load distribution towards the left limbs, whereas negative values describe a load distribution towards the right limbs. Values are given separately for left and right gait direction, e.g., 1.20/−0.40 denoting *LatAI* = +1.20% for left walk and *LatAI* = −0.40% for right walk in the forelimbs. Standard gravity is *g* = 9.81 ms^−2^.

## Data Availability

Data are contained within the article.

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
