# Peer review of "Applying Multi-Purpose Commercial Inertial Sensors for Monitoring Equine Locomotion in Equestrian Training"

_sensors, 2024, doi:10.3390/s24248170_

Round 1
Reviewer 1 Report (Previous Reviewer 1)
Comments and Suggestions for Authors
Thank you for revising and representing your research, it is now improved in terms of your intention versus the scientific method used. I have only noted a few suggestions for revisions:
Line 42 have not had
Line 44 full stop not colon
Line 166 why did you use 20 horses? Was a power calculation carried out - does this number adequately answer your validity question?
Line 175 Is this for one horse or averaged data for the 20 horses?
Line 176 error for figure 4
Figure 4 - legend should state it is the right lead canter
Discussion
Line 339 as you have not tested reliability, ie inter- horse or inter-horse repeated occasional please remove the suggestion that you have demonstrated reliability of your system. However I agree it appears to be able to monitor and provide data
Line 433 can you support your point with a reference to known data on the laterality/symmetry of clinically normal riding horses.
Author Response
Please see the attachment.

Reviewer 2 Report (Previous Reviewer 2)
Comments and Suggestions for Authors
General Remarks :
The article “Applying smart inertial sensors for exercise monitoring in equestrian sports” aims to propose a pilot study on the use of inertial sensors for monitoring the locomotion of the animal.
This revised article focuses on horse locomotion. However, as already mentioned in the first submission, the use of IMUs for the quantification of horse locomotion has already been extensively studied, and the demonstration of the feasibility of IMUs but also of the use in clinical practice has been carried out. In addition, the authors use methods already published. Consequently, the innovative part is also limited.
The authors announced in the title the realization of monitoring which would involve the analysis of the parameters studied on several training sessions or change of learning technique. That is not being done at all.
Finally, throughout the text, the authors refer to human locomotion, which is not at all appropriate because no comparison can be made between bipedalism and quadrupedalism.
In conclusion, the scientific relevance of this article in its current form is limited.
Specific remarks :
Line 2: the title must be adapted to the content of the text. In this new version it is only about horse locomotion, not equestrian sport.
Line 78-91: the link to human locomotion is not at all appropriate as there are already many studies using IMUs for horse locomotion analysis. Consequently, this paragraph provides no new information that would be relevant to the analysis of horse locomotion.
Line 92-98: the objective of the study has a limited impact as many have already demonstrated the relevance of inertial sensors for the horse locomotion study.
Line 120-122: limitations of the material used should be included in the section that details study limitations and potentially perspective possibilities
Line 128: again, the link to human locomotion is inappropriate
Line 174: the term ground contacts is ambiguous because it is here the determination of a temporal event.
Line 176: reference error.
Line 200: Specify the frequency of the video camera. In the case of a difference in frequency between the video frequency and the IMU frequency, you can estimate the error for identifying spatiotemporal events.
Line 220: it is not specified whether straight lines and turns were studied; This is important because it is known that the IMU signals are not the same in a straight line as in turns
Line 345: Xsens sensors are not known to be low cost; it is better to say here that these are IMU on the market.
Line 409-422: this paragraph is speculative, what parameters allow here to make the connection with the movements of the trunk?
Line 430: Again, the comparison between human locomotion is not appropriate. There are also studies that asymptomatic asymmetries for the horse.
Round 2
Reviewer 2 Report (Previous Reviewer 2)
Comments and Suggestions for Authors
I thank the authors for taking into account the remarks of the previous review and for proposing appropriate complement informations and corrections. The document is now consistent with agreed objectives and outcomes.
Below two minor points:
-Line 15, the word "inexpensive" can be removed as it is a subjective point of view.
-Line 435 the sentence here must be more precise that this is a future possibility; I therefore propose "This method could be in a further study easy to implement.... ”
Author Response
Please see the attachment.

This manuscript is a resubmission of an earlier submission. The following is a list of the peer review reports and author responses from that submission.
Round 1
Reviewer 1 Report
Comments and Suggestions for Authors
Thank you for submitting the report of your very in depth descriptive reporting of the IMU data collection of equine movement. Whilst I can see how much work this has been for you, I am unable to recommend publishing at this time. You methods do not have contain any testing of reliability, nor does it compare to a gold standard. Therefore there is no ability to ensure that the data output and methods for stride segmentation are accurate or valid. You could retitle and change you aims, noting this major limitation. My concern is that without validation this system could be marketed or used in research but without any understanding of its accuracy.
I appreciate my comments will be frustrating but a inclusion of a comparison to a gold standard is needed for the system to have value for future researchers and projects.
Reviewer 2 Report
Comments and Suggestions for Authors
The publication titled "Utilizing Smart Inertial Sensors for Motion Monitoring and Exercise Control in Elite Equestrian Sports" aims to serve as a pilot study investigating the application of inertial sensors in analyzing various equestrian activities. However, critical issues prevent its immediate publication:
1)The study's objective is to showcase the significance of employing inertial sensors for quantifying equestrian activity. While this theme has seen numerous publications and specific studies on horse biomechanics, the authors must clarify the innovative aspects of their research.
2)The paper covers several types of equestrian activities without establishing clear connections between them. Additionally, the number of subjects per study varies significantly, ranging from 4 to 20 subjects. It is recommended to focus on a specific activity, providing context on why inertial sensor measurements are necessary and the benefits they offer.
3)The authors frequently draw analogies between human and horse locomotion, which is inappropriate and irrelevant. Given the extensive literature on horse biomechanics, such comparisons are unnecessary. Furthermore, comparing horse locomotion to itself is nonsensical, rendering the analogy irrelevant.
4)Methodologically, the authors define cycles without demonstrating or validating whether they correspond to the same spatiotemporal events for forelimbs and hind limbs. This is particularly crucial as the sensors are placed on the cannon, introducing a time delay compared to direct hoof measurements. Without validating the coherence of space-time events between forelimbs and hind limbs, the use of the "longAI" parameter remains questionable.
5)The authors ambiguously address the potential saturation of measurements due to sensor limitations (±16 g). It is well-known that inertial sensors with a larger measurement range are required for activities like trotting, galloping, and jumping. Additionally, saturation may only occur on one axis, potentially unnoticed in acceleration amplitude. Furthermore, the impact of gravity on their analysis is overlooked.
6)In the analysis of jumps, the authors determine the trunk rotation angle without detailing the method or validating the measurements. Additionally, Figure 8 exhibits clear angular drift, a known limitation that is a strong limitation.
In conclusion, addressing the aforementioned issues necessitates significant restructuring of the publication. Further experiments are required to validate the reliability of the conducted measures. Analogies with human locomotion are irrelevant to the analysis of horse locomotion.